# Genetic Basis, New Diagnostic Approaches, and Updated Therapeutic Strategies of the Syndromic Aortic Diseases: Marfan, Loeys–Dietz, and Vascular Ehlers–Danlos Syndrome

**DOI:** 10.3390/ijerph20166615

**Published:** 2023-08-20

**Authors:** Laura Asta, Gianluca A. D’Angelo, Daniele Marinelli, Umberto Benedetto

**Affiliations:** 1Department of Cardiac Surgery, Tor Vergata University Hospital, 00133 Rome, Italy; 2Department of Cardiac Surgery, SS Annunziata Hospital, 66100 Chieti, Italy; gantonio.dangelo@asl2abruzzo.it (G.A.D.); daniele.marinelli@asl2abruzzo.it (D.M.); umberto.benedetto@asl2abruzzo.it (U.B.)

**Keywords:** aneurysm, dissection, Marfan syndrome, Loeys–Dietz syndrome, vascular Ehlers–Danlos syndrome, syndromic aortic diseases

## Abstract

Syndromic aortic diseases (SADs) encompass various pathological manifestations affecting the aorta caused by known genetic factors, such as aneurysms, dissections, and ruptures. However, the genetic mutation underlying aortic pathology also gives rise to clinical manifestations affecting other vessels and systems. As a consequence, the main syndromes currently identified as Marfan, Loeys–Dietz, and vascular Ehlers–Danlos are characterized by a complex clinical picture. In this contribution, we provide an overview of the genetic mutations currently identified in order to have a better understanding of the pathogenic mechanisms. Moreover, an update is presented on the basis of the most recent diagnostic criteria, which enable an early diagnosis. Finally, therapeutic strategies are proposed with the goal of improving the rates of patient survival and the quality of life of those affected by these SADs.

## 1. Introduction

Syndromic aortic diseases (SADs) are congenital diseases that encompass aortic manifestations such as aneurysm formation, dissection, and aortic rupture, which can also occur in non-dilated vessels. These conditions (according to genetic disorders) are often just one aspect of a more complex clinical picture. The underlying genetic mutations of SADs can also give rise to manifestations in other body systems.

When a specific genetic mutation associated with a particular clinical and physical phenotype can be identified, we refer to syndromic aortopathies as Marfan syndrome (MFS), Loeys–Dietz syndrome (LDS), and vascular Ehlers–Danlos syndrome (vEDS). On the other hand, we refer to non-syndromic aortopathies when aortic manifestations result from gene mutations that change components of vascular smooth muscle cells but are not associated with other systemic abnormalities. However, it is not always possible to identify the specific genetic mutation responsible for these conditions at present.

The genetic basis plays a key role in defining not only the best diagnostic program but also the most fitting medical therapy or surgical treatment.

The aim of this review is threefold. First, we summarize the current genetic evidence on the basis of SADs. Second, we describe the pathophysiological processes in order to define the new diagnostic approaches currently recognized. Finally, we try to define the time and type of the surgery or, alternatively, when there is no indication for surgery, the most appropriate medical treatment in accordance with the latest guidelines.

## 2. Genetic Basis and Pathophysiology

Recent advancements in the study of DNA sequencing led to the identification of a large series of genes responsible for the alterations of the great vessels’ connective tissue.

These breakthroughs have significantly contributed to our understanding of the underlying pathophysiology and have unveiled potential therapeutic targets [1].

The major pathways genetically implicated in the development of SADs include proteins involved in the formation of:−Extracellular matrix (ECM);−vascular smooth muscle cells (VSMCs);−transforming growth factor-beta (TGF-β) [2].

The ClinGen Aortopathy Working Group has put forth a systematic classification of the genes whose variation predisposes to thoracic aortic disease.

The classification consists of a scale based on how strong the link is between genetic variation and aortic diseases (Figure 1) [3].

### 2.1. Marfan Syndrome

MFS is an autosomal dominant syndrome of the connective tissue that affects the cardiovascular, skeletal, ocular, pulmonary, and nervous systems, with a prevalence of one in 5000/10,000 persons [4]. MFS is caused by mutations in the FBN1 gene that encodes fibrillin-1, a major component of the extracellular matrix [5].

The FBN1 also interferes in the down-regulation of TGF-β (transforming growth factor-β) through TGF-β cytokines (large latent complexes (LLC) that contain latency-associated peptide (LAP) and latent TGF-β binding protein (LTBP) anchored to the extracellular matrix with fibrillin-1). In the presence of mutated fibrillin-1, the altered pathway leads to an overexpression of TGF-β (Figure 2) [6].

TGF-β binding to receptors activates the SMAD-dependent canonical cascade. SMAD-independent noncanonical pathways are also activated, and the activation of extracellular signal-regulated kinases (ERK1/2) becomes an important driver of aortic aneurysm formation (Figure 3) [7]. In several studies, it has been shown that angiotensin II (AngII) type 1 receptor blockers (ARBs), such as losartan, have a protective role in aneurysmal formation by preventing noncanonical ERK1/2 activation [8].

Mutations in TGF-β signaling pathway-related genes play a decisive role in the pathogenesis of MFS because these genetic alterations lead to structural changes in the connective structure of the vessels, which facilitate dilatation and dissection [6]. In the aortic wall of patients with MFS, it has been seen fragmentation and disorganization of elastic fibers in the media layer, calcification, and thickening of all layers due to the accumulation of collagen and amorphous matrix, in addition to focal intimal thickening in all vessels [9]. Additionally, Radonic et al. have demonstrated how this altered collagen structure leads to the synthesis of degradation products that attract inflammatory cells. The inflammatory process, in turn, aggravates disease severity in patients with MFS [10].

### 2.2. Loeys-Dietz Syndrome

Loeys–Dietz syndrome (LDS) is an autosomal dominant connective disorder caused by mutations in different members of the TGF-β pathway.

In order to give an account of the different mutations, a new classification was put forward in 2014 that defines LDS variants in IV types. Initially, LDS was classified into two types, depending on the severity of craniofacial features (type 1) or skin features (type 2). However, evidence that certain genetic mutations were more associated with the onset of aortic aneurysmal disease or dissection led to a categorization based on those genetic mutations that were deemed responsible [11].

In the aftermath of this classification, two other forms of LDS were proposed: type 5 associated with the *TGFB3* mutation and type 6 determined by the *SMAD2* mutation.

Bertoli-Avella et al. analyzed 11 families with a history of syndromic aortic aneurysms. The analysis showed that in 43 subjects, there was a correlation with *TGFB3* mutations. A strong correlation in human aortic tissue between *TGFB3* loss-of-function (LOS) mutations and enhancement of the TGF-β signaling pathway was highlighted. Furthermore, in subjects affected by this mutation, not only cardiovascular involvement (thoracic/abdominal aortic aneurysm and dissection, mitral valve disease) but also clinical features typical of LDS (hypertelorism, bifid uvula, cleft palate) were present [12,13].

Gene mutations of *TGFBR1* and *TGFBR2* result in overexpression of the TGF-β pathway, resulting in increased connective tissue growth factor (CTGF) and increased nuclear accumulation of phosphorylated *SMAD2* (pSMAD2) [14]. *SMAD2*, after its phosphorylation, has a fundamental role in the control of gene transcription; therefore, its altered functioning determines an alteration of the cellular signal. Mutations affecting the *SMAD2* gene (especially the missense and nonsense variants) are related to the most recently identified type of LDS, LDS type 6. Micha et al. were among the first to associate the *SMAD2* mutation with a new form of LDS, starting from the analysis of three families with a history of aneurysm and dissection in which there was evidence of overexpression of *SMAD2* at the level of the aortic wall [15].

It has also been observed that the *SMAD3* protein also plays a key role in the activation/repression of the TGF-β pathway. In particular, truncating mutations that delete nearly the complete MH2 domain in *SMAD3* lead to an increase in the activity of TGF-β [16].

Thus, immunohistochemical evidence of increased expression of TGF-*β* receptors, *TGFBR2* and *TGFBR3*, as well as intracellular downstream effectors of the TGF-*β* pathway, *SMAD2* and *SMAD3*, in the aortic media of LDS patients demonstrates that the TGF-*β* pathway has a crucial role in the pathogenesis of aortic aneurysm and dissection [17].

In fact, the overexpression of the TGF-β pathway in SMCs leads to a loss of basal *SMAD* signaling, decreased expression and activity of contractile molecules, and altered stress-related signaling [18]. Initially, the histopathologic features of MFS and LDS appear to overlap. Actually, it was then seen that in LDS, diffuse media degeneration is more represented, i.e., increased medial collagen, a diffuse form of elastic fiber fragmentation, and extracellular matrix deposition [19].

### 2.3. Vascular Ehlers-Danlos Syndrome

Ehlers–Danlos syndromes (EDS) are a heterogeneous group of hereditary connective tissue disorders characterized by extreme clinical and genetic variability. Initially, six main types of EDS were classified by Villefranche Nosology (1997) [20]. But in 2017, the International EDS Consortium proposed a revised EDS classification that recognizes 13 subtypes. The classification is based on the correlation between the gene and the corresponding phenotype. Furthermore, for each of the subtypes, a set of clinical criteria suggestive of the diagnosis has been proposed (Figure 4) [21].

The vascular Ehlers–Danlos syndrome (vEDS) is an autosomal dominant disorder that results from mutations in the *COL3A1* gene, which encodes the pro-alpha1 chain of type III procollagen [22], affecting approximately 1 in 50,000 to 250,000 people [23].

Just under 700 (exactly 670) mutations regarding COL3A1 have been identified: Missense substitutions affecting one of the glycine residues of the [Gly-X-Y]343 repeat within the triple helical region of type III collagen (the majority), splice-site variants of exons encoding a triple helix sequence leading to an in-frame exon skipping and generation of a shorted translated product also altering the stable assembly of type III procollagen homotrimers, rare frameshifts, non-sense, small or large deletions (that result in a reduction of mature procollagen and haplo-insufficiency) [24].

Therefore, the different genetic mutations that lead to a destabilization of type III procollagen are responsible for the different degrees of phenotypes. In particular, a more or less precocious onset of major events (vascular, digestive, or obstetric) has been highlighted based on the type of genetic mutation affecting *COL3A1* (Figure 5) [25].

Usually, glycine missense and splice-site mutations (mutations called dominant negative, DN) are the variants associated with an earlier and more severe course of the disease. Furthermore, it has been seen that patients with COL3A1 DN mutations have a greater risk of aneurysms, dissections, and ruptures of medium-sized vessels (such as the carotid arteries) compared to the aorta [26].

The aberrant III pro-collagen causes fragility of the vessels, with a consequent lower resistance to wall stress and a higher risk of spontaneous ruptures. At the histological level, in the mouse model, a progressive fragmentation of the tonic intima and of the media was highlighted, with sparing of the adventitia, leading to the proliferation of smoothelin-positive medial vascular smooth muscle cells as well as fibroblasts and the production of a collagenous scar [27].

## 3. Diagnostic Approach

The diagnosis of SADs relies on three fundamental elements: obtaining a thorough medical history, conducting a comprehensive physical examination, and performing genetic screening.

Clinical criteria play a significant role in characterizing specific syndromes and facilitating the diagnostic process. In particular, the presence of aortic aneurysmal disease in young individuals can serve as an initial indication that prompts consideration of genetic counseling.

The American Heart Association proposes a protocol for the genetic study of patients with thoracic aortic disease presenting syndromic features, a family history of thoracic aortic disease (TAD), and/or an early age of disease onset. If the genetic test is positive, it will be necessary to proceed with the genetic study of the first-degree relatives. However, if the genetic test is negative or reveals variants of unknown significance (VUS), the family members will have to undergo instrumental examinations (ETT, CT, MRI) for the study of the aorta (Figure 6) [28].

Among the various diagnostic techniques, magnetic resonance imaging (MRI) plays a leading role in the assessment and follow-up of aortic dilatation thanks to the accuracy of the data it can provide but also to the safety of the method in young patients (absence of ionizing radiation), pregnant women, or in cases of chronic kidney disease as there is no need for intravenous contrast. Furthermore, the evolution of 4D-flow provides, through the study of blood flow encoded in three-dimensional velocity, numerous hemodynamic parameters, including wall shear stress (WSS), pulse wave velocity (PWV), kinetic energy, and turbulent kinetic energy (TKE), which allow a comprehensive analysis of complex blood flow patterns. Therefore, MRI can be considered a risk stratification tool [29,30].

From a surgical point of view, thanks to ECG triggering and therefore processing a study synchronized with the heart rate, it is possible to minimize movement artifacts and obtain highly detailed images of aortic anomalies as well as anatomical relationships with contiguous anatomical structures. For this purpose, the introduction of magnetic resonance angiography (MRA) has further increased the definition of the spatial relationships between the structures of the mediastinum. The use of paramagnetic contrast medium within the vascular system increases the contrast between the vessel and surrounding anatomical structures, regardless of flow and velocity patterns, thereby reducing pulsatility artifacts [31].

### 3.1. Marfan Syndrome

In the context of such a complex syndrome due to clinical variability, the age-dependent nature of many of its manifestations, the absence of gold standards, and its extensive differential diagnosis, the modified Ghent criteria are proposed as more stringent than the previous classifications (such as Berlin nosology) and providing better guidelines to differentiate MFS from related, “overlapping” conditions (Figure 7) [32].

The Ghent criteria include clinical manifestations defined as major and minor involving several systems (skeletal, ocular, cardiovascular, and pulmonary systems, as well as the dura, skin, and integument). The diagnosis of MFS is made in the presence of:−major involvement of at least two organ systems, with minor involvement of a third organ system;−major and one minor manifestation in different organ systems (in addition to the FBN1 mutation known to cause MFS or a first-degree relative who was unequivocally diagnosed based upon Ghent nosology).

The most common cardiovascular manifestations are dilatation of the aorta and mitral regurgitation [33].

The dilatation in MFS affects the sinuses of Valsalva and the tubular portion of the ascending aorta, giving the typical shape defined as “pear-shaped” [2]. The risk of aortic dissection (the leading cause of death in patients with MFS [34]) has been shown to be greater in the presence of aortic tubular dilation than in the presence of root dilation alone [35]. Furthermore, the *FBN1* truncating mutation is associated with a higher probability of developing aortic events and with an earlier age than the missense mutation [36].

Therefore, any patient, whether a child or adult, being evaluated for MFS should undergo an echocardiogram (Figure 8) [37]. However, at the time of diagnosis, in addition to TTE, a global evaluation of the aorta (arc, descending aorta, and abdominal aorta by CT or MRI) is useful.

Furthermore, since the growth rate of the aorta in patients with MFS is higher than in others, continuous instrumental evaluation (echo, CT, and MRI) is essential over time [38,39]. If the diameter of the aorta remains stable below 45 mm, annual checks can be performed; if the aorta is larger than 45 mm, more frequent checks are recommended [40]. If TTE is not technically feasible (e.g., due to chest malformations), an MRI evaluation is recommended to reduce the radiation rate and ensure greater protection of the kidneys.

### 3.2. Loeys-Dietz Syndrome

At present, there are no precise criteria for the diagnosis of LDS. Therefore, the diagnosis is based on the genetic analysis associated with the phenotypic study and family history.

The latest classification of LDS includes six different types (associated with specific genetic mutations) [24]:−type I (*TGFBR1* mutation) with the following clinical manifestations: cleft palate, craniosynostosis, micrognathia, and/or bifid uvula; rare valve abnormalities; arterial tortuosity; ascending aortic aneurysm and dissection; pulmonary artery aneurysm;−type II (*TGFBR2* mutation), also called vascular EDS-like LDS, with a clinical picture characterized by aortic aneurysms, arterial tortuosity, hypertelorism, abnormal uvula, joint laxity, pectus deformity, scoliosis, and arachnodactyly [41];−type III (*SMAD3* mutation) known as aneurysm-osteoarthritis syndrome, which includes: early-onset joint abnormalities such as osteoarthritis and osteochondritis dissecans; aneurysms and tortuosity of the aorta and other arteries throughout the body, including intracranial arteries; mitral valve prolapse and regurgitation; aortic insufficiency; left ventricular hypertrophy; atrial fibrillation; varices; spider veins [42];−type IV (*TGFB2* mutation) with a high prevalence of aortic aneurysms, mitral valve prolapse, aortic root aneurysm and dissection, arterial tortuosity, cerebrovascular aneurysm and tortuosity, joint laxity, pectus deformities, and scoliosis [41];−type V (*TGFB3* mutation) provides the typical clinical features of LDS, but a clear correlation with the risk of dissection at a young age has not been highlighted, and furthermore, the arterial anomalies are not characterized by the typical vascular tortuosity detectable in other forms of LDS;−type VI (*SMAD2* mutation), two different phenotypic models have been proposed. The first foresees a purely cardiac involvement (valvular anomalies, dextrocardia), and the second includes vascular manifestations such as aneurysms and dissections of the thoracic and abdominal aorta (Table 1) [43].

Whereas in MFS, aortic involvement mainly affects the aortic root and ascending aorta, in LDS there is more involvement of the epiaortic and cerebral vessels and mesenteric arteries (Figure 9) [42].

Furthermore, aneurysmal pathology in LDS is characterized by greater tortuosity of the vessels and higher fragility of the walls.

### 3.3. vEhlers–Danlos Syndrome

Diagnosing vEDS is not straightforward, as many clinical features may be common to other disorders [23]. vEDS can be diagnosed when two of four diagnostic criteria (thin, translucent skin; arterial, intestinal, or uterine rupture; easy bruising; and a characteristic facial appearance) are met together with confirmation of the responsible genetic mutation. The diagnosis should be considered and a biochemical evaluation performed in young people with unexplained bowel or arterial rupture, especially those with a family history of similar events [44].

Therefore, the most important vascular impairment of vEDS is the rupture of small-medium-caliber vessels (such as renal, iliac, femoral, hepatic arteries, etc.) at a young age.

In the event that there is aortic involvement (aneurysms or dissections), this is associated with a high mortality rate determined by the extreme tissue fragility and by the high rate of complications (bleeding, sternal wound dehiscence) in the immediate perioperative time (Figure 10) [45,46].

## 4. Therapeutic Strategies

### 4.1. Surgical Treatment

The surgical indication for SADs is based on the size of the aneurysm, its growth rate, and, therefore, the risk of rupture.

However, SADs comprise a heterogeneous group of genetic and phenotypic conditions. Therefore, in the surgical decision, various elements must be evaluated, such as the localization and site of the aneurysm, the presence of aortic or mitral regurgitation, but also the clinical condition, the pathogenic variant, a personal or family history of dissection, or a desire for pregnancy [47].

According to ESC (European Society of Cardiology) Guidelines, in MFS, patients should undergo surgery when the aortic root maximal diameter is ≥50 mm, or 46–50 mm, but with a family history of dissection, progressive dilation >2 mm/year as confirmed by repeated measurement, severe AR or MR, or a desire for pregnancy [48]. Instead, in LDS, the ESC recommends a more aggressive approach, suggesting cardiac surgical treatment at a diameter >42 mm due to the extreme tissue fragility of this genetic disorder. In vEDS, there is not enough scientific evidence to establish a cut-off diameter [49].

There are no significant differences between the guidelines of the American Heart Association (AHA) [28].

The high clinical variability of SADs means that the surgical strategy, both open and endovascular, must be personalized for the individual patient based on the clinical condition, comorbidities, and associated risks [50].

Current techniques include replacement of the ascending aorta with aortic root reconstruction (remodeling or reimplantation), Bentall operation, arch replacement with reimplantation of epiaortic vessels, debranching of epiaortic vessels, or the hybrid procedure called frozen elephant trunk (FET) (Figure 11).

Considering the genetic etiology and the high reintervention rate these patients experience, the use of a hybrid technique such as FET allows for easier second-stage operations, providing a platform for surgical and endovascular reinterventions [51]. Furthermore, it has been shown that even in the aortic dissection of MFS patients, FET determines a positive remodeling of the distal aorta, ensuring an increase in survival and freedom from reoperation in the long term [52].

However, the rate of peri- and post-operative complications is significantly higher than in non-syndromic patients. In fact, although the average age is lower, the underlying connective disorder predisposes to a greater risk of respiratory failure, bleeding, and shock [53], without significant differences between teaching and non-teaching hospitals in terms of in-hospital mortality, overall morbidity, length of stay, or total hospital charges [54].

### 4.2. Endovascular Repair

Because of the tortuosity of the vessels and the congenital tendency for aneurysmal dilatation, endovascular treatment (TEVAR) is not the first choice in SADs, as the risk of endoleak is significantly high [55], as is the risk of reoperation [56].

However, since it is a percutaneous treatment, the related risks are considerably lower than with the open technique (low perioperative mortality, spinal cord ischemia, or cerebrovascular accident) [57]. Therefore, TEVAR finds its greatest application in the event of an emergency, in life-saving procedures in which immediate treatment and a low rate of complications must be guaranteed.

### 4.3. Medical Therapy

The goal of medical therapy in SADs is to reduce hemodynamic stress but, given the altered pathways in the individual genetic syndromes, also reduce the remodeling processes affecting the aortic wall and the consequent process of dilatation.

Therefore, the drugs studied are those that pharmacologically target the TGF-β pathway, such as β-blockers and losartan [58].

According to the latest American guidelines, β-blockers and sartans continue to be the drugs of first choice in the treatment of SADs [28]. β-blocker drugs cause a decrease in heart rate and reduce the rate of aortic growth. Losartan, an angiotensin II receptor antagonist, reduces the phosphorylation of *SMAD2* and inhibits the ERK-2 kinase pathway, resulting in a negative regulation of TGF-β [59].

Although no differences in the rate of aortic dilatation or in the frequency of clinical events were identified between the two drug groups, the use of losartan is associated with a lower risk in the long-term management of these patients [60].

In the treatment of vEDS, the protective role of celiprolol is now established. Celiprolol is a cardioselective β-blocker with β2 agonist vasodilatory characteristics; therefore, by reducing the pressure, it causes a reduction of the arterial wall stress, preventing the risk of aneurysms and/or dissections. Furthermore, a β3 adrenoceptor agonistic role of celiprolol has been highlighted, which allows for the activation of endothelium- and nitric oxide-dependent pathways [61]. Ong et al. demonstrated in their trial that the group treated with celiprolol (compared to the control group) had a lower risk of cardiovascular events [62]. More recently, Frank et al., through a long-term observational study (average follow-up of 5 years), confirmed the increase in the survival rate in patients treated with celiprolol (especially if administered at a dosage of 400 mg/day) [63].

## 5. Management in Pregnancy

The physiological, hormonal, and hemodynamic changes of pregnancy significantly prolong the risks of aortic rupture not only during the gestational period but also in the postpartum period [64]. Therefore, women with hereditary heart disease who wish to become pregnant should receive a multidisciplinary evaluation as well as counseling regarding the increased risks during pregnancy [65]. Furthermore, during pregnancy, continuous instrumental follow-up, via ETT or MRI, is necessary for close monitoring of the aortic diameters.

## 6. Prognosis

The life expectancy of patients affected by SADs has certainly undergone a significant improvement in the last 20 years thanks to ever earlier genetic diagnoses, more defined follow-up programs, and therefore the possibility of performing elective surgery. The event that most negatively affects the prognosis is the occurrence of aortic dissections.

Robertson et al. showed in their study of the outcomes of patients with SADs that aortic dissection was present in every group of genetic disorders analyzed, particularly in MFS (especially type B dissection as a complication after surgery on the ascending aorta) [66].

However, as emerged from GenTAC (genetically triggered thoracic aortic aneurysm and cardiovascular conditions) data, aortic surgery in patients with MFS is associated with excellent outcomes [67].

In LDS, on the other hand, it has been seen that some factors (*TGFBR2* mutation, female gender, aortic tortuosity, hypertelorism, and translucent skin) are associated with a worse prognosis, understood as a greater risk of aortic dissection, and therefore, in such circumstances, there is an indication for more timely surgical treatment [68].

Observational studies available on vEDS are still quite limited due to very small cohorts. However, it emerges from the GenTAC that in these patients, mortality from cardiovascular causes is higher, probably due to the greater aggressiveness of the disease [67].

## 7. Conclusions

SADs are genetic disorders characterized by extreme complexity both in diagnosis and clinical management. The identification of the responsible genetic mutation is the key element to understanding the responsible pathogenic path determined by the altered pathway on the one hand, and to guaranteeing the patient the best treatment on the other. Furthermore, once a diagnosis has been reached, it is crucial to prioritize genetic counseling for both the patient and the family. The counseling serves to ensure that the patient receives all the necessary information on the risks associated with the diagnosed genetic disorder. The ongoing advancements in medical therapy and in the improvement of surgical techniques and the consequent reduction in peri- and post-operative risks will allow for the guarantee of acceptable expectations and quality of life for patients.

## Figures and Tables

**Figure 1 ijerph-20-06615-f001:**
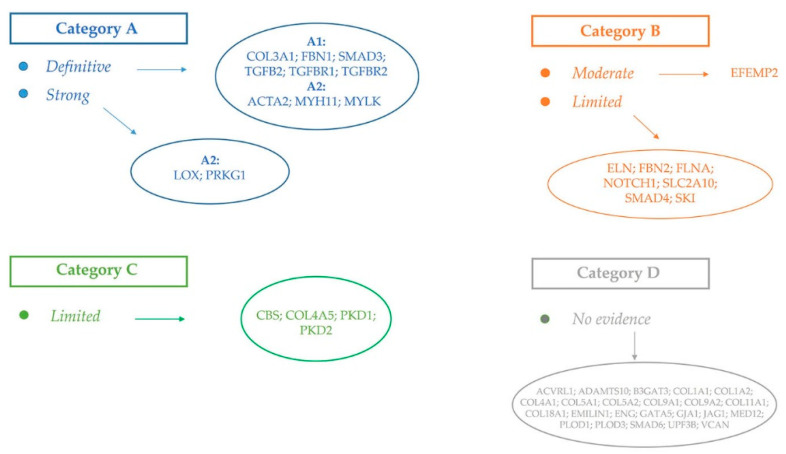
Classification of the genes responsible for SADs. Modified by [3].

**Figure 2 ijerph-20-06615-f002:**
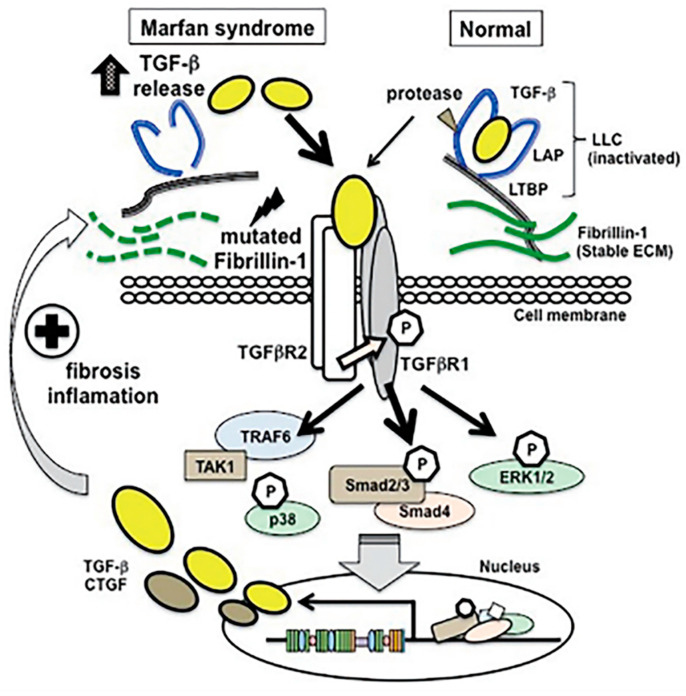
*FBN1* mutation. Mutation of the *FBN1* gene causes a dysregulation of the TGF-β pathway, leading to increased activation. Adapted with permission from [6].

**Figure 3 ijerph-20-06615-f003:**
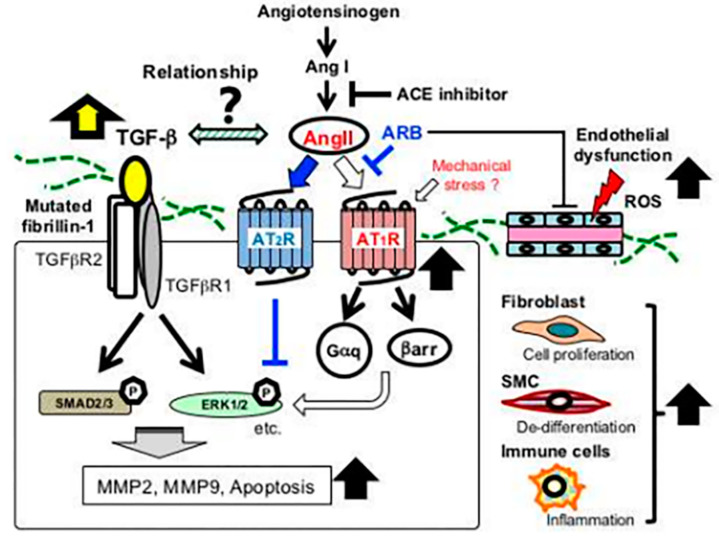
FBN1 mutation and aortic aneurysms. The overexpression of metalloproteinases (MMP2; MMP9) and pro-apoptotic factors, causing a structural alteration of the smooth muscle cells (SMCs), determines the formation of aortic aneurysms. Adapted with permission from [7].

**Figure 4 ijerph-20-06615-f004:**
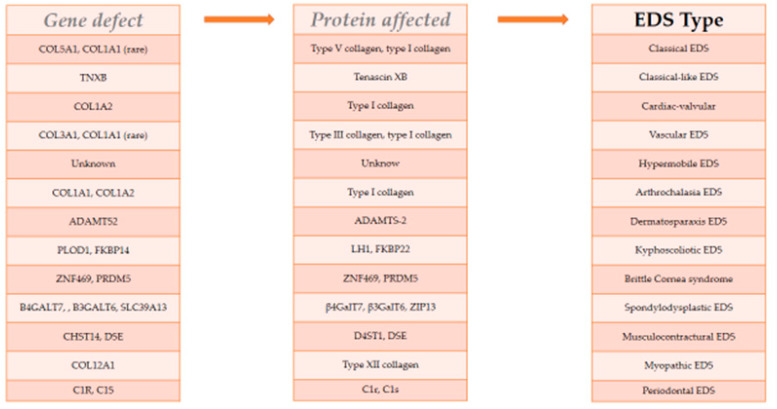
EDS 2017 Classification. The correlation between the gene defect and the affected protein determines the type of EDS. Modified by [21].

**Figure 5 ijerph-20-06615-f005:**
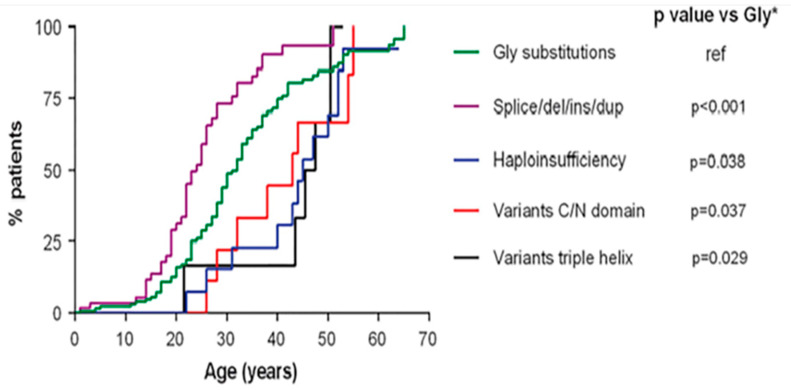
Age of onset of first major complication in relation to COL3A1 variant type. * Glycine. Adapted with permission from [25].

**Figure 6 ijerph-20-06615-f006:**
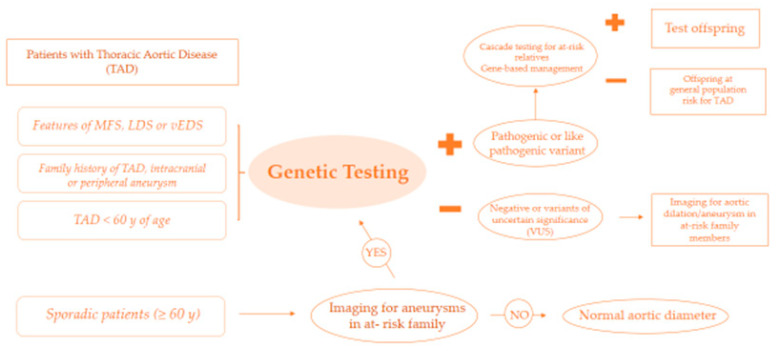
Protocol for the genetic study. Modified by [28].

**Figure 7 ijerph-20-06615-f007:**
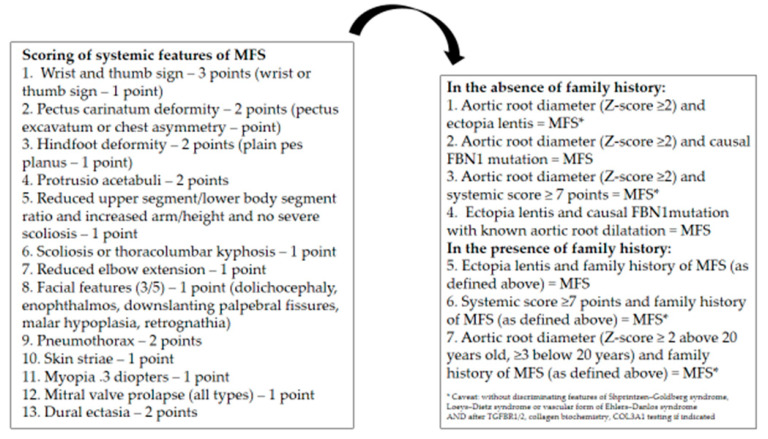
Revised Ghent criteria and scoring of systemic features. Modified by [32].

**Figure 8 ijerph-20-06615-f008:**
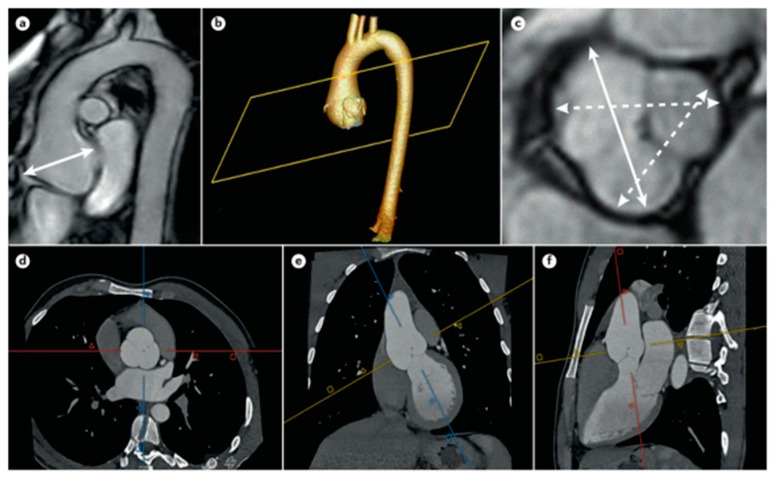
Aortic aneurysm in MFS. (**a**,**c**) Aortic root aneurysm by MRI measured in sagittal projection and transversal projection, evaluating cusp to cusp diameters at end-diastole; (**b**) 3D reconstruction of CTA imaging (**d**–**f**) of an aortic root aneurysm; the double acquisition of sagittal and coronal images allows for a more correct transversal diameter of the aortic lumen. Adapted with permission from [37].

**Figure 9 ijerph-20-06615-f009:**
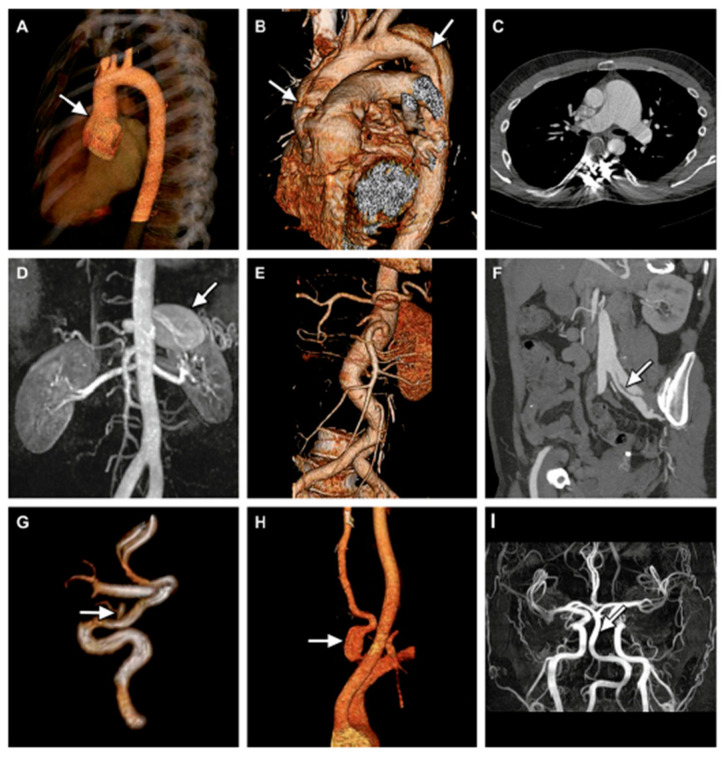
Arterial anomalies in LDS. (**A**): Aortic root aneurysm by 3D CT angiography (CTA); (**B**) Stanford type A aortic dissection extending into the brachiocephalic trunk; (**C**) aneurysm of the truncus pulmonalis by CT; (**D**) aneurysm of the splenic artery by RMI; (**E**) tortuosity of the abdominal aorta, suprarenal aneurysm of the abdominal aorta and aneurysms of the coeliac trunk, and left common iliac artery by 3D CTA; (**F**) Stanford type B aortic dissection of abdominal aortic with dissection flap extending into the left common iliac artery; (**G**) saccular aneurysm of the right ophthalmic artery by RMI; (**H**) fusiform aneurysm of the left vertebral artery by 3D CTA; (**I**) fusiform dilatation of basilar artery by RMI. Adapted with permission from [42].

**Figure 10 ijerph-20-06615-f010:**
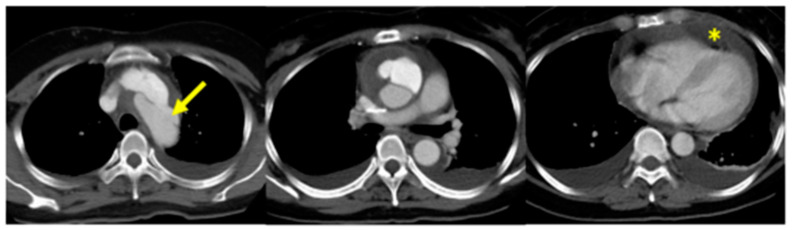
Stanford type A dissection in patient with vEDS. Arrow: intimal tear in the aortic arch; asterisk: pericardial effusion. Adapted with permission from [46].

**Figure 11 ijerph-20-06615-f011:**
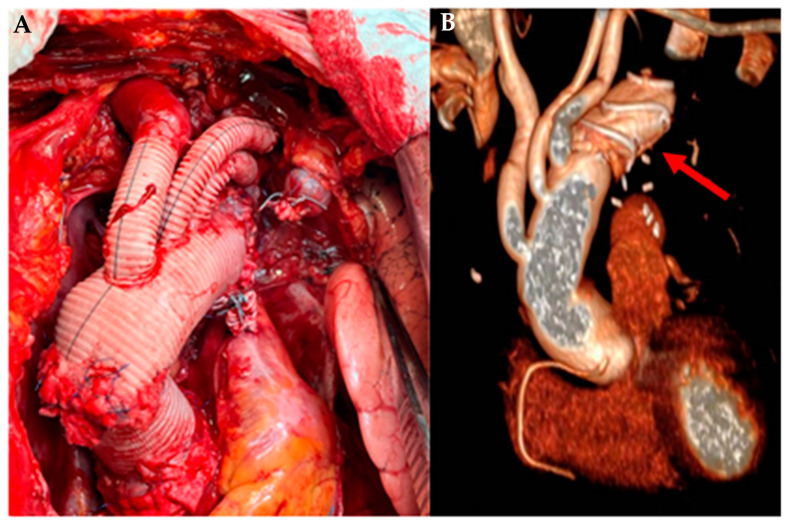
FET technique in SS Annunziata Hospital. (**A**) Intraoperative image of aortic root and aortic arch replacement with single reimplantation of the epiaortic vessels; (**B**) image processed by 3D CT showing (arrow) the stent grafting in the descending thoracic aorta.

**Table 1 ijerph-20-06615-t001:** LDS 2023 Classification.

LDS Type	Responsible Gene	Cardiovascular Manifestation
LDS 1	*TGFBR1*	Valve abnormalities (rare): bicuspid aortic valve, bicuspid pulmonary valve, mitral valve prolapse; arterial tortuosity; ascending aortic aneurysm and dissection; pulmonary artery aneurysm
LDS 2	*TGFBR2*	Valve abnormalities (rare); arterial tortuosity; ascending aortic aneurysm and dissection; pulmonary artery aneurysm
LDS 3	*SMAD3*	Valve abnormalities: mitral valve prolapse and regurgitation, aortic insufficiency; left ventricular hypertrophy; atrial fibrillation; aortic aneurysm and dissection; arterial aneurysm and tortuosity; varices; spider veins
LDS 4	*TGFB2*	Mitral valve prolapse; aortic root aneurysm; aortic dissection; arterial tortuosity; cerebrovascular aneurysm and tortuosity
LDS 5	*TGFB3*	Mitral and aortic insufficiency; aortic root dilation; aneurysm/dissection of thoracic and abdominal aorta; elastic fiber fragmentation observed in the aneurysmal aortic wall; varices
LDS 6	*SMAD2*	Valve prolapse or insufficiency (aortic, mitral, tricuspid, or pulmonary valve); thoracic or abdominal aortic aneurysm; arterial and aorta tortuosity; dextrocardia

## Data Availability

Not applicable.

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
