# Peer review of "Genetic Basis, New Diagnostic Approaches, and Updated Therapeutic Strategies of the Syndromic Aortic Diseases: Marfan, Loeys–Dietz, and Vascular Ehlers–Danlos Syndrome"

_ijerph, 2023, doi:10.3390/ijerph20166615_

Round 1

Reviewer 1 Report

Remarks on the manuscript entitled "Heritable Aortopathies (Marfan, Loeys Dietz and vEhlers Danlos syndrome): genetic basis, new diagnostic approaches and therapeutic strategies”.

Unfortunately, the manuscript is neither up-to-date (e.g., the LDS classification is incomplete and the medical therapy of vEDS does not consider celiprolol) nor worded in a way that would allow me to review it with reasonable effort. Thank you for your understanding.

Appropriate editing of English language required.

Author Response

Dear Reviewer,

thank you for your time.

I'm sorry for your negative feedback. We have therefore tried to expand our manuscript with your suggestions. In particular, we have expanded the classification of Loeys-Dietz syndrome both in the paragraph concerning the genetic basis and pathophysiology (2.2) as well as in the paragraph concerning the diagnostic approach (3.2). We have also added a section on the use of celiprolol in the medical treatment of vEDS. You find all the extra parts written in red.

I hope our insights may have improved the quality of our manuscript.

Reviewer 2 Report

Dear Authors

I would like to thank you for the opportunity to review this interesting paper focused on a very remarkable and challenging topic that is a lively argument also in daily clinical practice. 

The aim of the present review is to summarize the current knowledge on the genetic basis, the pathophysiological processes, the diagnostic approaches and the updated therapeutic strategies of the main heritable aortopathies, including Marfan syndrome, Loeys Dietz syndrome and Ehlers Danlos syndrome.

This paper is pleasurable to read, although it suffers from some limitations that Authors can easily adjust to slightly improve their review making it more eligible for this important Journal. Furthermore, the Authors can improve some sections of the paper, adding information and including other important references about this topic that, in my opinion, should be cited and discussed. 

First of all, I believe there is a typo in the title (“vEhlers”) that should be corrected. Moreover, from a merely stylistic point of view, I suggest to rearrange the title to a catchier version: “Genetic basis, new diagnostic approaches and updated therapeutic strategies of the syndromic aortic diseases: Marfan syndrome, Loeys Dietz syndrome and Ehlers Danlos syndrome”.

Secondly, although the language used is quite appropriate, I (I am not a native English speaker) recommend to the Authors obtain a certified native speaker with proficiencies in the scientific-medical field to complete properly this paper and check for spelling errors (for example, line 28 “genetic mutansion”).

Lines 37-41, please add also “new diagnostic approaches” as one of the main goals of the review.

Paragraph 2: please avoid the extensive use of lists since the diagram reported in Figure 1 is sufficient to schematize the classification.

Please check all the tables and stick to the same font and dimension, otherwise is difficult to read them; also, keep abbreviations to a minimum.

Figures, when possible, should be new and not adapted from other works. For example, Figure 5 is a simple scheme that is very simple to replicate with the modifications suggested by the Authors. Please correct as many Figures as possible.

Please expand the role of imaging in both diagnosis and follow up of patients with hereditable aortopathies and try to explain which imaging modality is preferable. For example, magnetic resonance imaging (MRI) almost certainly offers the greatest advantages, especially in young patients in which radiation exposure must be avoided as much as possible. MRI provides excellent visualization of vascular structures with a wide field of view, well suited for evaluation of the thoracic aorta malformations. With the implementation of magnetic resonance angiography (MRA) it is also possible to depict any relationship with supra-aortic or mediastinal vessels [doi: 10.1007/s00330-005-0027-y][doi: 10.1053/j.ro.2022.07.004]. Please discuss this topic and cite the aforementioned references.

Finally please add a brief paragraph regarding the prognosis and the possible long-term complications associated with these diseases.[ PMID: 20301299] [doi: 10.3389/fcvm.2022.1009947]

I think references should be reformatted as suggested by the Author’s guidelines (Author 1, A.B.; Author 2, C.D. Title of the article. Abbreviated Journal Name YearVolume, page range).

Best regards,

Author Response

Dear Reviewer,
Thank you for your time.
We took your advice and modified our manuscript as follows:
- vEhlers in the title, it was referred to the vascular type of Ehlers-Danlos syndrome. However, we thought the title you proposed was more suitable, as you can see in the new version.
- the text has been reviewed by a certified native speaker who has corrected some errors and reworked some sentences in a more correct way
- we have added "new diagnostic approaches" as one of the main objectives of the review.
- the description relating to figure 1 has been eliminated
- all tables and stick have the same character and size
- figure 5 has not been adapted from the referenced article, but has been replicated with modifications by the authors
- we integrated the role of imaging in both diagnosis and follow-up with reference to MRI and MRA (citing the references you proposed) and making a brief mention of the use of 4D-Flow.
- we have added a brief paragraph on the prognosis and possible associated long-term complications
- we have reformatted the references as suggested by the Author's guidelines.
You find all the extra parts written in red.

I hope our insights may have improved the quality of our manuscript.

Reviewer 3 Report

 From a teaching point of view,  It is an interesting review about  aortopathies it but it does not contribute anything new   . 

Authors should explain whether they have obtained permission to adapt the figures. There are some errors like " Heler Danlos " that should be corrected. Section 3.3 should be reviewed (authors say LDS but they are talking about Ehlers Danlos I think)

English language seems correct but as I am not native , it would be approppriate to revise it carefully 

Author Response

Dear Reviewer

Thank you for your time.

The aim of our review was to take stock of the new genetic, diagnostic and therapeutic knowledge of syndromic aortic diseases, proposing itself as an information tool for greater awareness of genetic aortopathies in the scientific field.

We have inserted under each figure the permission obtained for the adaptation (fig 4 has been removed since it was not possible to obtain the permission for the re-sharing of the image).

The errors mentioned have been corrected and furthermore the text has been reviewed by a certified native speaker who has corrected some errors and reworked some sentences in a more correct way.You find all the extra parts written in red.

I hope our insights may have improved the quality of our manuscript.

Round 2

Reviewer 1 Report

Remarks on the manuscript entitled "Genetic basis, new diagnostic approaches and updated therapeutic strategies of the syndromic aortic diseases: Marfan, Loeys Dietz and Ehlers Danlos syndrome”.

The authors have revised the manuscript (R1), but in my opinion, there are still several points that need to be improved, e.g.
- there are several repetitions (redundancy) in the manuscript (may consider chapters/points 2, 3, and 4 per syndrome);
- Abstract: first sentence needs revision since “caused by precise genetic factors” is misleading;
- avoid adjectives such as “extremely”;
- consider that EDS is not equal to vEDS => revise in the entire manuscript;
- line 33: consider that SAD are also caused by a “single gene mutation”;
- line 48: revise sentence;
- Figure 1 is adapted from Reference [3] but this was not indicated accordingly;
- “FBN-1” => “FBN1” and use it consistently in the entire manuscript (also consider that genes should be written in italic)
- revise “Syndrome” vs “syndrome” in the entire manuscript;
- Table 1 from 2014 is outdated in 2023 (cf. www.omim.org/phenotypicSeries/PS609192);
- Table 1: “Clinical manifestation” is incomplete (may consider moving details from the text here);
- line 138: revise “truncating a mutation”;
- lines 140-141: “TGFBR2 and TGFBR3“ are receptors and NOT ligands;
- line 158: consider “gene” instead of “genetic mutation”;
- lines 170-175: What is the source for more than 700 COL3A1 mutations (HGMD Pro, ClinVar)? Consider that “heterozygous” applies to all COL3A1 mutation types leading to vEDS (autosomal dominant);
- lines 421-428: also consider PMID: 33223285 and double check “>” for 400 mg/day;
- double check references (cf. 41 and 43 are identical!);
- revise the title of the manuscript as well.

In my opinion, (moderate) editing of English language required.

Author Response

Dear Reviewer,

thank you for your time and suggestions.

We have addressed points posed by you as follows:

  • Abstract: we have replaced "precise" with "known" to indicate SADs caused by identified genetic mutations.
  • We have removed the adjective “extremely” where present
  • We have replaced EDS with vEDS in the entire manuscript since that is the subtype we are referring to
  • Line 33: we have expanded the definition of non-syndromic aortopathies: they too, like SAD, can be determined by a single genetic mutation but do not involve the involvement of other systems.
  • In figure 1 we have inserted the reference [3].
  • FBN-1 has been replaced with FBN1 and all gene names italicized
  • "Syndrome" has been revised to "syndrome"
  • Line 138: “truncating a mutation” was fixed to "truncanting mutation "
  • Lines 140-141: “TGFBR2 and TGFBR3“ have been fixed in "receptors"
  • Line 158: “genetic mutation” replaced with "gene"
  • Table 1: we have updated the classification to 2023 and expanded the clinic section by referring only to cardiovascular manifestations in line with the review topic
  • Lines 170-175: we have corrected the mutation number (just under 700, 670) and inserted the reference (HGMD), eliminating the wording "heterozygous" referring to the splicing mutation since, as correctly suggested, they are all heterozygous mutations.
  • Lines 421-428: we added the suggested reference (PMID: 33223285) on the role of celiprolol on the β3 receptor and we corrected the dosage of celiprolol as we meant that the mentioned article (Frank et al) demonstrated that the role of celiprolol is dependent dosage and that the maximum result is obtained for a dosage of not less than 400 mg/day.
  • We performed a double check references
  • Line 48: In what terms should we revise the sentence (form, language)?
  • The title of the review has already been changed as suggested by another reviewer. However we have specified the type (vascular) of Ehlers Danlos to which we refer
  • We revised the paragraphs and tried to eliminate some repetition.

Reviewer 2 Report

The authors addressed raised points adequately.

Author Response

thank you

Reviewer 3 Report

With respect to the previous manuscript,  the authors have improved both the text and the writing including the improvements suggested by the reviewers

Author Response

thank you